# *AaHog1* Regulates Infective Structural Differentiation Mediated by Physicochemical Signals from Pear Fruit Cuticular Wax, Stress Response, and *Alternaria alternata* Pathogenicity

**DOI:** 10.3390/jof8030266

**Published:** 2022-03-06

**Authors:** Miao Zhang, Tiaolan Wang, Yongcai Li, Yang Bi, Rong Li, Jing Yuan, Wenyi Xu, Dov Prusky

**Affiliations:** 1College of Food Science and Engineering, Gansu Agricultural University, Lanzhou 730070, China; zhangmiao202202@126.com (M.Z.); wangtiaolan@hotmail.com (T.W.); biyang@gsau.edu.cn (Y.B.); lir_gsau@163.com (R.L.); yj15214029875@163.com (J.Y.); xuwenyi0813@163.com (W.X.); dovprusk@volcani.agri.gov.il (D.P.); 2Department of Postharvest Science of Fresh Produce, Agricultural Research Organization, The Volcani Center, Rishon LeZion 50250, Israel

**Keywords:** *Alternaria alternata*, *AaHog1*, pear fruit cuticular wax, stress response

## Abstract

The high-osmolarity glycerol response kinase, *Hog1*, affects several cellular responses, but the precise regulatory role of the *Hog1* mitogen-activated protein (MAP) kinase in the differentiation of the infective structure of *Alternaria*
*alternata* induced by pear cuticular wax and hydrophobicity has not yet clarified. In this study, the *AaHog1* in *A. alternata* was identified and functionally characterized. *AaHog1* has threonine-glycine-tyrosine (TGY) phosphorylation sites. Moreover, the expression level of *AaHog1* was significantly upregulated during the stages of appressorium formation of *A. alternata* on the fruit-wax-extract-coated GelBond hydrophobic film surface. Importantly, our results showed that the appressorium and infection hyphae formation rates were significantly reduced in Δ*AaHog1* mutants. Furthermore, *AaHog1* is beneficial for the growth and development, stress tolerance, virulence, and cell-wall-degrading enzyme activity of *A. alternata*. These findings may be useful for dissecting the *AaHog1* regulatory mechanism in relation to the pathogenesis of *A. alternata*.

## 1. Introduction

Black spots caused by the necrotrophic fungi *Alternaria alternata* is the most serious postharvest disease of pear fruit [1]. During the early stage of infection, *A. alternata* conidia adhere to the surface of pear fruits, where conidial spores gradually germinate into germ tubes. Upon recognizing suitable environmental signals, the tip of the germ tubes swell and form the appressorium, and partial appressorium differentiates under enormous turgor pressure into infectious hyphae [2,3,4]. However, the underlying regulatory mechanisms of *A. alternata* sensing on the pear fruit surface and infection initiation is poorly understood.

The physicochemical signals from the plant surface include hydrophobicity, surface hardness, ethylene, waxes, and cutin [5]. The plant cuticle, a membranous structure on the aerial epidermis of all land plants, plays a dual role of plant self-protection and inducing plant pathogenic fungal infection [6]. Increasing evidence shows that plant cuticular wax is a key inducer of appressorium-formation in several fungi, such as *Magnaporthe oryzae*, *Puccinia graminis*, *A. alternate*, and *Colletotrichum gloeosporioide* [2,7,8,9]. Feng et al. [10] suggested that the wax monomer alkanes C24 and C25 significantly promote *Blumeria graminis* spore germination. Two compounds, 12-hydroxystearic and 16-hydroxyhexadecanoic acid, strongly induce differentiation of the *Ustilago maydis* infectious structure [11]. However, an important stimulus signal, hydrophobicity, induced appressorium formation in *M. oryzae* and *B. graminis* [5,9]. Furthermore, the plant hormone ethylene affected the conidial germination of *Botrytis cinerea* [12]. In *M. grisea* and *C. gloeosporioides*, a hard hydrophobic surface is required for appressorium formation [13]. However, there is limited information on pathogenic fungal recognition, the response in the form of physicochemical cues of wax from the fruit surface, and then the initiation of infection.

The G protein-coupled receptor recognizes and responds to various exogenous signals, which are then transmitted to mitogen-activated protein (MAP) kinase cascades or cyclic AMP-dependent protein kinase (cAMP-PKA) signal pathways to trigger diverse cellular processes [14,15,16]. MAP kinases belong to the serine/threonine-specific protein kinase family and are highly conserved. There are three different types of MAP kinase in most pathogenic fungi, which are the orthologs of the model fungi *Saccharomyces cerevisiae Fus3*/*Kss1*, *Slt2* and *Hog1* MAP kinase. Evidence proved that *Fus3*/*Kss1, Slt2,* and *Hog1* MAP kinases are critical for regulating growth, infection structure formation, cell wall integrity, and high osmolarity response in plant-pathogenic ascomycetes [17]. Alonso et al. [18] reported that loss of *Hog1* increases the intracellular reactive oxygen content in *Candida albicans*. Besides, *ClHog1* and *BoHog1* are also indispensable for virulence in *Colletotrichum lagenarium* and *Bipolaris oryzae* [19,20]. The *Fusarium graminearum Fghog1* mutant strain is abnormal during corn and wheat infection [21], but in *Christella parasitica*, deleting the *Hog1* ortholog *Cpmk1* causes melanin deposition and weakens virulence against the host [22]. Additionally, deleting the *Hog1* mutant in *Fusarium proliferatum*, *Aspergillus nidulan,* and *C. lagenarium* significantly increased tolerance to phenylpyrrole and dicarboximide fungicides [23,24].

The effect of the *Hog1*-MAP kinase has been identified and analyzed in many fungi, but the precise regulatory role of *AaHog1* in the differentiation of the infection structure of *A. alternata* remains unknown. This study assessed the function of *AaHog1* during the appressorium and infection hyphae formation of *A. alternata* mediated by physicochemical cues of pear fruit cuticular wax. The study also investigated the regulatory role of *AaHog1* on the stress response, pathogenicity, and the cell-wall-degrading enzyme activity of *A. alternata*. These findings may be useful for dissecting the regulatory mechanism of *AaHog1* MAP kinase in the pathogenesis of *A. alternata.*

## 2. Materials and Methods

### 2.1. Fungal Strains

The wild-type (WT) strain (KY397985.1) of *Alternaria alternata* was isolated from naturally decayed ‘Zaosu’ pear fruit and identified by sequencing the 16 S and aligning to the *A. alterna**ta* genome. All the strains used in this experiment were grown on potato dextrose agar (PDA) media at 28 °C.

### 2.2. Construction of Plasmid Vectors for ΔAaHog1 and AaHog1-c

The *AaHog1* (XM_018530379.1) deletion mutant was constructed using the homologous recombination strategy (Appendix A) to verify the function of *AaHog1* in *A. alternata*. Briefly, 5′ (1 kb) and 3′ (1 kb) untranslated regions (UTRs) of *AaHog**1* were amplified and ligated sequentially to flank the pCHPH vector using the ClonExpress II one-step cloning kit (Vazyme Biotech Co., Ltd., Najin, China) following the manufacturer’s instructions. The vector plasmids were confirmed by sequencing and subsequently transformed into WT Agrobacterium tumefaciens to replace the *AaHog1* gene [25]. Second-round transformants were screened using 80 µg mL^−1^ hygromycin B (Roche, Mannheim, Germany), and specific primer pairs verified the presence of the transformant band. Thus, it was verified that *AaHog1* was completely knocked out (Appendix A). The correct transformants were further confirmed using quantitative RT-PCR (qRT-PCR). The amplified fragments and qRT-PCR primers are shown in Appendix A.

The *AaHog1* complementation strain (AaHog1-c) was constructed in reference to a previously described method [26]; a 1.3 Kb DNA fragment containing the entire AaHog1 gene as well as the promoter region *Olic* and terminator region *Trpc* were amplified using Q5 high-fidelity DNA polymerase (GIBCO, Wolcavi Beijing Biological Technology co., LTD, Beijing, China). The DNA fragment was inserted at the *Xbal*I and *Sac*I site of pC-NEO-CGFP (provided by the Chinese Academy of Sciences). The final plasmid was then transformed into Δ*AaHog1* mutant using the PEG-mediated transformation method. Transformants were picked from PDA media containing 250 μg/mL G418 (GIBCO, Wolcavi Beijing Biological Technology co., LTD, Beijing, China) for the second selection round. The correct transformants were further confirmed by amplification using specific primer pairs. Appendix A shows the amplified fragments.

### 2.3. Quantitative RT-PCR Analysis

Spore suspensions (5 × 10^5^ spores/mL) of WT strains were placed onto a hydrophobic film and fruit wax-coated hydrophobic film surface. Total RNA was extracted from *A. alternata* at different time points after incubation using the TRIzol reagent (QIAGEN, Shanghai, China) following the manufacturer’s protocol. Approximately 2 µg of the RNA was reverse transcribed to cDNA for qRT-PCR analysis (Takara Biological Technology co., LTD, Dalian, China). *GAPDH* was the reference gene. Each sample had three qRT-PCR replicates. The Livak and Schmittgen [27] method was adopted to calculate the relative gene expression. The qRT-PCR primers are shown in Appendix A.

### 2.4. Extraction of Fruit Wax

Fruit wax was extracted using a previously described method [2]. Briefly, ‘Zaosu’ pear fruits were put in 600 mL chloroform and agitated for 60 s at room temperature. The extracting solution was filtered with eight layers of gauze, and the solvent was removed by vacuum distillation. The dried pear cuticular wax extract was refrigerated at 4 °C for subsequent experiments.

### 2.5. Infection Structure Formation Assays

The GelBond hydrophilic and hydrophobic film (Youningwei Biotechnology Co., Ltd., Shanghai, China) was cut into 5 cm × 2 cm rectangles. The hydrophilic and hydrophobic films were placed on clean glass slides as the first treatment group. Then, the hydrophobic film was coated with either 20 μL paraffin wax, 40 μL fruit wax, or 60 μL beeswax and placed on a clean glass slide as the second treatment group. Then, 20 μL conidial suspensions (1 × 10^5^ spores/mL) from the WT strain, Δ*AaHog1* mutant, or AaHog1-c strain were pipetted onto the treated hydrophobic film, respectively. All assays were performed in triplicate. Spore germination and appressorium formation rates were calculated at 2, 4, 6, and 8 h after incubation, at 28 °C and 95% relative humidity [28].

The intact, dewaxed, fruit-wax-extract-coated, and fruit-wax-extract-coated dewaxed onion epidermises were placed on clean glass slides. Then, 20 μL conidial suspensions (1 × 10^5^ spores/mL) from the WT strain, Δ*AaHog1* mutant, or AaHog1-c strains were pipetted onto the treated onion epidermis, respectively. All assays were performed in triplicates. Appressorium and infecting hyphae formation rates were calculated 2, 4, 6, and 8 h after incubation at 28 °C and 95% relative humidity.

### 2.6. Growth and Development Phenotype Analysis

Two-microliter conidial suspension (1 × 10^5^ spores/mL) of WT, Δ*AaHog1* mutant, and AaHog1-c strains were inoculated on the center of PDA plates covered with sterile cellophane films under sterile conditions. Then, the colony and cellophane films were removed and weighed seven days after incubation. Each strain had three replicates.

Colony growth and sporulation were detected following previously described methods [29]. The PDA medium was inoculated with 2 μL conidial suspensions (1 × 10^5^ spores/mL) of WT, Δ*AaHog1* mutant, or AaHog1-c strains and inoculated at 28 °C. Photographs were taken, and the colony diameters were measured daily. Spores from each medium were collected using 10 mL of sterile water after seven days of incubation. A hemacytometer was used to count the spore concentration, and assays were performed in triplicates.

### 2.7. Stress Response

Stress response was determined following a previously described method [25]. Conidial suspensions were inoculated on PDA media containing different compounds (1 mol L^−1^ NaCl, 1 mol L^−1^ sorbitol for osmotic stress; 10 μmol^−1^ Congo red, and 1 mol^−1^ sodium dodecyl sulfate (SDS) for cell wall integrity), with the PDA medium serving as the control. The assays were analyzed in triplicate. Pictures were taken, and colony diameter was measured six days after incubation. The percentage of inhibition = (P − S)/S × 100%, where P denotes the colony diameter in the PDA medium, and S denotes the colony diameter in WT, Δ*AaHog1* mutant, and AaHog1-c strains in response to exogenous stresses.

### 2.8. Virulence Determination

Pathogenicity was analyzed based on a previously described method with minor modifications [30]. Briefly, ‘Zaosu’ pears were washed and air-dried at room temperature. Then, 20 μL of conidial suspensions (1 × 10^6^ spores m L^−1^) from WT, Δ*AaHog1* mutant, or AaHog1-c strains were inoculated at each wound (2 mm deep, 5 mm wide) of the ‘Zaosu’ pear fruit. Each strain was used to inoculate nine pear fruits, and the inoculated fruits were packed in plastic boxes and stored at 23 °C under 55% humidity. The lesion diameter of ‘Zaosu’ pears was measured 1, 3, 5, and 7 days after treatment. The assays were performed in triplicates.

### 2.9. Cell-Wall-Degrading Enzyme Activity Analysis

Following Chiu et al. [31], crude enzyme solutions were extracted with minor modifications. Basically, 10 mL conidial suspensions (1 × 10^6^ spores mL^−1^) of WT, Δ*AaHog1* mutant, or AaHog1-c strains were inoculated in 50 mL Erlenmeyer flasks containing 15 mL potato dextrose media, respectively. The inoculated suspensions were cultured in the dark at 28 °C and 120 rpm for four days. Next, the hyphae were transferred to culture solutions containing pectin and cellulose and cultured in the dark for 0–8 days before use. The hyphae were removed by filtration, and the filtrate was centrifuged at 4 °C and 12,000 r/min for 20 min. The precipitate was discarded, and the supernatant was retained as the crude enzyme solution for determining enzyme activity.

Polygalacturonase (PG) and pectin methylesterase (PMG) activities were determined following the protocol of Yoshida et al. [32]. The test tube contained 0.4 mL citrus pectin (PG) or 1% polygalacturonic acid (PMG), 0.1 mL enzyme extract, and 0.5 mL sodium acetate buffer (4.5 pH and ice-cooled). After incubation at 37 °C for 1 h, 1 mL of dinitrosalicylic acid (1%) was added to the reaction mixture and heated in a boiling water bath for 5 min. The mixture was measured immediately at 540 nm. One enzyme activity unit (U) was considered as the amount of enzyme that needed to be released per μg of reduced sugar from the per-mg quantity of enzyme protein per hour. The PG and PMG activities were expressed as U mg^−1^ of protein.

Then, 1.5 mL of 10 g/L substrates (Cellulase: sodium carboxymethyl cellulose, β-glucosidase: salicin) were added to test tubes (Cellulase: sodium carboxymethyl cellulose, β-glucosidase: salicin) and incubated for 5 min at 37 °C. The crude enzyme solution (0.5 mL) was added to one of the test tubes, and the inactivated enzyme was added to the other test tube as the control. The mixture was incubated in a 37 °C water bath for 60 min. Immediately after incubation, 1.5 mL DNS reagent was added to the mixture and further incubated in a boiling water bath for 5 min, then quickly cooled to room temperature. The absorbance values were determined at 540 nm following the DNS method. Moreover, cellulase (Cx) and β-glucosidase activities were expressed as U mg^−1^ protein. One enzyme activity unit (U) was described as the amount of enzyme required to catalyze substrate (sodium carboxymethylcellulose and salicin) hydrolysis per hour and the per-mg reduction in sugars.

### 2.10. Statistical Analysis

The average and standard error (±SE) of the data were calculated in Microsoft Excel 2010. Significance analysis with Duncan’s multiple range test was performed at a 5% level using SPSS 19.0 (SPSS Inc., Chicago, IL, USA).

## 3. Results

### 3.1. Characterization and Identification of the AaHog1-MAP Kinase

*AaHog1* was identified via cloning from the *A. alternata* (txid5599) genome. Sequence analysis showed that *AaHog1* genomic DNA contained a 1615 bp open reading frame and seven small introns: 49, 49, 48, 47, 49, 46, and 46 bp (Appendix A). The Phylogenetic analysis indicated that *AaHog1* and *Hog1* of *A. longipes* (ANZ79239.1) belong to the same evolutionary branch, with approximately 74% homology (Appendix A). In addition, *AaHog1* encoded a 355 amino acid polypeptide that is highly conserved among fungi. The *AaHog1* amino acid sequence deduced using the GENEDOC software aligned with the *Hog1* MAPK sequences of other fungi. Moreover, *AaHog1* possessed a TGY phosphorylation conserved domain, with characteristic sequences of the Ser/Thr protein kinase active site VvHRDLKPSNILv and ATP-binding region GMGAFGLVCSAkDq*T*QVAvKKi (Figure 1a).

### 3.2. Gene Expression

Gene expression results revealed that *AaHog1* was remarkably upregulated during the early stage of appressorium formation on the GelBond hydrophobic film surface coated with fruit wax extract (4 h after incubation). Compared with the spore germination stage (2 h after incubation), the *AaHog1* relative gene expression level was upregulated 12 fold on the GelBond hydrophobic film coated with fruit wax extract at 4 h post-incubation (Figure 1b).

### 3.3. AaHog1-MAPK Is Important for Infective Structure Formation Induced by Physicochemical Cues from the Pear Fruit Surface

As shown in Figure 2, the GelBond hydrophilic (θ1) and hydrophobic (θ2) films promoted *A. alternata* spore germination, while the hydrophobic film surface dramatically induced appressorium formation. The Δ*AaHog1* mutant significantly decreased the rates of spore germination and appressorium formation (*p* < 0.05). Besides, the spore germination and appressorium formation rates of the Δ*AaHog1* mutant on hydrophobic film surfaces were reduced by 11% and 94% after 4 h of incubation, contrary to the WT strains. Additionally, the infective structure formation rate of the AaHog1-c was comparable to the WT strain (Figure 2a,b).

The spore germination and appressorium formation rates of *A. alternata* on fruit wax (F) coated surface were dramatically induced when compared to the paraffin (P)-, and beeswax (B)-coated surfaces (Figure 2c,d). The strain carrying a defective *AaHog1* locus displayed significantly low germination and appressorium formation rates (17% and 67%) compared to the WT strain after 4 h of incubation on the fruit wax surface (*p* < 0.05). Moreover, the defects in infective structure formation were rescued in the AaHog1-c strain.

The effect of *AaHog1* on *A. alternata* infection hyphae formation was also evaluated on the surface of intact (θ3), dewaxed (θ4), fruit-wax-extract-coated (θ5), and fruit-wax-extract-coated dewaxed (θ6) onion epidermises. As shown in Figure 2, the fruit-wax-extract-coated dewaxed onion epidermis surface significantly promoted *A. alternata* appressorium and infection hyphae formation (Figure 2e,f). Similarly, the infection hyphae formation rate of the Δ*AaHog1* mutant on the different onion epidermises was lower than the WT and AaHog1-c strains.

### 3.4. AaHog1 Is Important for A. alternata Growth, Development and Stress Response

The Δ*AaHog1* mutant strain severely retarded growth and caused irregular margins in the colony (Figure 3a). As anticipated, deleting *AaHog1* reduced *A. alternata* biomass and conidia production by 23% and 13% as compared to the WT strain (Figure 3b,c). However, the AaHog1-c strain partially restored the defect of Δ*AaHog1* mutant in sporulation.

Next, we analyzed whether *AaHog1* is involved in different *A. alternata* stress responses. The results showed that treatment with 1 mol L^−1^ NaCl almost completely inhibited Δ*AaHog1* growth, and the colony appeared as round dots (Figure 4). The percentage of Δ*AaHog1* inhibition was markedly higher than the WT and AaHog1-c strains. Similarly, the percentage of WT and AaHog1-c inhibition was −32.08 and −28.42% in sorbitol media, but the Δ*AaHog1* mutant strain showed 66.76% inhibition (Table 1). When cultured on PDA media with Congo red (CR) and sodium dodecyl sulfate (SDS), the Δ*AaHog1* mutant was more sensitive to cell wall inhibitors than the WT and AaHog1-c strains.

### 3.5. AaHog1 Is Indispensable for A. alternata Virulence

Fungal virulence was investigated by means of wound inoculation of ‘Zaosu’ pear fruit using conidial suspensions of the WT, Δ*AaHog1* mutant, and AaHog1-c strains. The disease spot area on the ‘Zaosu’ pear fruit inoculated with the Δ*AaHog1* mutant was remarkably lower than in fruits inoculated with the WT strain. At seven days after inoculation, the lesion diameter in pear fruits inoculated with the Δ*AaHog1* mutant was reduced by 51% and 50% as compared to fruits inoculated with the WT and AaHog1-c strains, respectively (Figure 5).

### 3.6. AaHog1 Positive Regulates Cell-Wall-Degrading Enzyme Activity

As shown in Figure 6, the PG activity first increased in the WT strain and gradually declined during the nine days of incubation. The PG activity significantly decreased (98%) in the Δ*AaHog1* mutant five days after cultivation compared to the WT strain. A similar pattern was observed in PMG, Gx, and β-glucosidase activities in the strain carrying the defective *AaHog1* locus compared to WT and AaHog1-c strains. In addition, the PG, PMG, Cx, and β-glucosidase activities of the AaHog1-c stain reverted.

## 4. Discussion

The mitogen-activated protein (MAP) kinase belongs to the serine/threonine-specific protein kinase family conserved in many fungi [17]. In this study, the high-osmolarity glycerol response kinase, *AaHog1*, was identified and functionally characterized in *A. alternata*. The results showed that the deduced amino sequence of *AaHog1* contains the unique phosphorylation motif “TGY” required for hyperosmotic response and a MAPK-active site characteristic sequence “VvHRDLKPSNILv” (Figure 1a). Thus, it was speculated that *AaHog1* probably has conserved functions in *A. alternata* growth, development, and/or pathogenicity.

The wax chemical composition and hydrophobicity of the host plant surface are important stimulus signals for inducing infection structure differentiation in phytopathogenic fungi [16]. The data presented in this study showed that the hydrophobic film and the surface coated with pear fruit wax extract significantly promoted *A. alternata* appressorium formation as compared to the hydrophilic surface (Figure 2). However, partial wax components such as normal alkanes, primary alcohols, fatty acids, and alkyl esters were not crucial for *B. graminis* pre-infection [33]. These results suggest that polar and non-polar components of cuticular wax have different inductive effects on the spore germination and appressorium formation of pathogenic fungi. Previous reports confirmed that the MAP kinase signal transduction pathway is important for the pathogenicity of all plant pathogenic fungi, especially for the formation of infective structures and the successful colonization of host tissues [34]. As anticipated, the Δ*AaHog1* mutant significantly suppressed the rates of spore germination and appressorium formation on different wax-coated surfaces compared to the WT and AaHog1-c strains.

Additionally, deleting *AaHog1* reduced the infective hyphae on the different onion epidermises, indicating that *AaHog1* regulated two critical steps of *A. alternata* pathogenesis: infection-structure formation and host-plant invasion. Joubert et al. [35] reported that the Δ*AbHog1* mutant strain delayed the conidial germination rate of *A. brassicicola*. In *B. cinerea* and *Meloidogyne graminicola*, mutants with a blocked *Hog1*-pathway are defective in plant infection [36]. On the contrary, *Osm1/Hog* was not important for appressorium formation in *M. oryzae*, and *Hog1* orthologs are optional for *Cochliobolus orbiculare* and *B. oryzae* infection [37]. These results showed that the *Hog1*-MAP kinase regulation of infective structure varies with the plant fungus.

*Hog1* MAP kinase is important for the growth and development of several fungi, including *F. graminearum* and *Verticillium dahlia* [36], but *Hog1* also regulates perithecium and microsclerotium formation. In this study, the biomass and sporulation of Δ*AaHog1* mutants were significantly low compared to the WT and AaHog1-c strains, indicating that *AaHog1* is required for *A. alternata* growth and development. In general, the “TGY” MAP kinase is indispensable for hyperosmotic stress. In *M. grisea* and *F**. graminearum*, the deletions of *Hog1* mutants are highly sensitive to osmotic stress [38,39]. Unsurprisingly, the Δ*AaHog1* mutant decreased tolerance to osmotic stress. This result also validated that the deletion of *AaHog1* in citrus *A. alternata* increases tolerance to osmotic stress [40]. Congo red and sodium lauryl sulfate (SDS) destroy the fungal cell wall by interfering with the cell wall construction and destroying cell wall proteins [41]. In this study, deleting *AaHog1* made *A. alternata* more sensitive to Congo red and SDS stress, implying that *AaHog1* is important for cell wall integrity. Interestingly, deleting *Fghog1* partially rescued the defects of the *mgv1* (ortholog of yeast *SLT2*) mutant for *F. graminearum* cell wall integrity [39]. Therefore, further studies to characterize the role of *AaSLT2* and *AaFus3* relationships with *AaHog1* in *A. alternata* are crucial.

The *Hog1*-pathway is thought to influence extracellular hyperosmotic stress, but subsequent studies have shown that Hog1-MAP kinase is also important for adaptation to abiotic/fungicide stress, secondary metabolism, and pathogenicity in filamentous fungi [40]. The pathogenicity assays in this study showed that the Δ*AaHog1* mutant had a dramatically decreased ability to infect pear fruits, similar to the effects of the Hog1-MAP kinase on the pathogenicity of citrus *A. alternata*, *C. lagenarium*, *B. oryzae*, *F. graminearum,* and *C. heterostrophus* [19,20,21,40,42]. Interestingly, the pathogenicity of the *Cshog1* mutant of *C**. sativus* was normal in the roots but significantly reduced in barley leaves [43]. These results suggest that the *Hog1*-MAP kinase’s regulatory role in virulence is specific for different fungal species and host tissues.

Filamentous fungal infections can cause softening and decay of plants or fruits due to the action of cell-wall-degrading enzymes, including PG, PMG, Cx, and β-glucosidase, with this being the main pathogenic factor of plant pathogens [44]. In this study, deleting *AaHog1* decreased PG, PMG, Cx, and β-glucosidase activities in *A. alternata* (Figure 6), indicating that *AaHog1* positively regulates the activity of PG, PMG, Cx, and β-glucosidase. These results confirmed that partially reducing the virulence of Δ*AaHog1* mutants on pear fruit reduced the activity of cell-wall-degrading enzymes. Additionally, the tentoxin content of Δ*AaHog1* mutants increased (results not shown) in *F. graminearum* and *Ustilaginoidea virens*, but the lack of *Hog1* reduced the production of phytotoxic metabolites [37,45]. Therefore, *AaHog1* possibly regulates other key pathogenic factors of *A. alternata*, but this hypothesis needs further evaluation.

In conclusion, *AaHog1* is involved in the vegetative growth, development, stress tolerance, virulence, and cell-wall-degrading enzyme activity of *A. alternata*. Importantly, this study provides new insights on the precise regulatory role of *Hog1*-MAP kinase in the differentiation of the infective structure of *A. alternata* induced by physicochemical signals from the pear fruit cuticular wax. A better understanding of the fine and collaborative regulatory mechanisms of MAPK and other signaling pathways would provide more information on the appressorium formation mechanisms of *A. alternata*.

## Figures and Tables

**Figure 1 jof-08-00266-f001:**
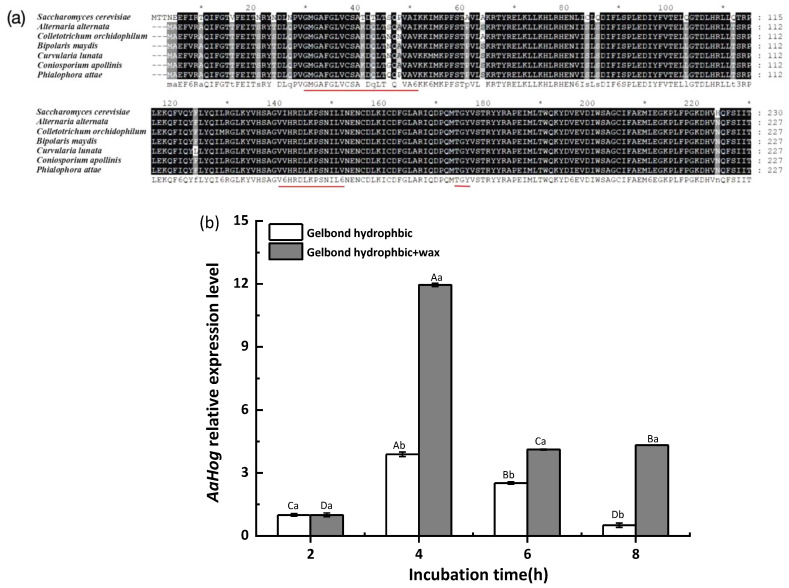
Characterization and identification of *AaHog1* in *A. alternata*. (**a**) Amino acid sequence alignments of AaHog1 with Hog1 MAPK of *Saccharomyces cerevisiae*, *Colletotrichum orchidophilum*, *Bipolaris maydis*, *Curvularia lunata*, *Coniosporium apollinis*, and *Phialophora attae* using the GENEDOC software. (**b**) The AaHog1 gene expression on the hydrophobic and hydrophobic + wax surfaces at spore germination (2 h), appressorium formation (4 h), germ tube elongation (6 h), and infection hyphae formation (8 h) stages. GAPDH was used as a reference gene, and assays were performed in triplicate. The bars represent the mean ± SD of three replicate samples. According to Duncan’s multiple range test, uppercase letters indicate significant differences between different times; lowercase letters indicate the significant differences between hydrophobic and hydrophobic + wax (*p* < 0.05).

**Figure 2 jof-08-00266-f002:**
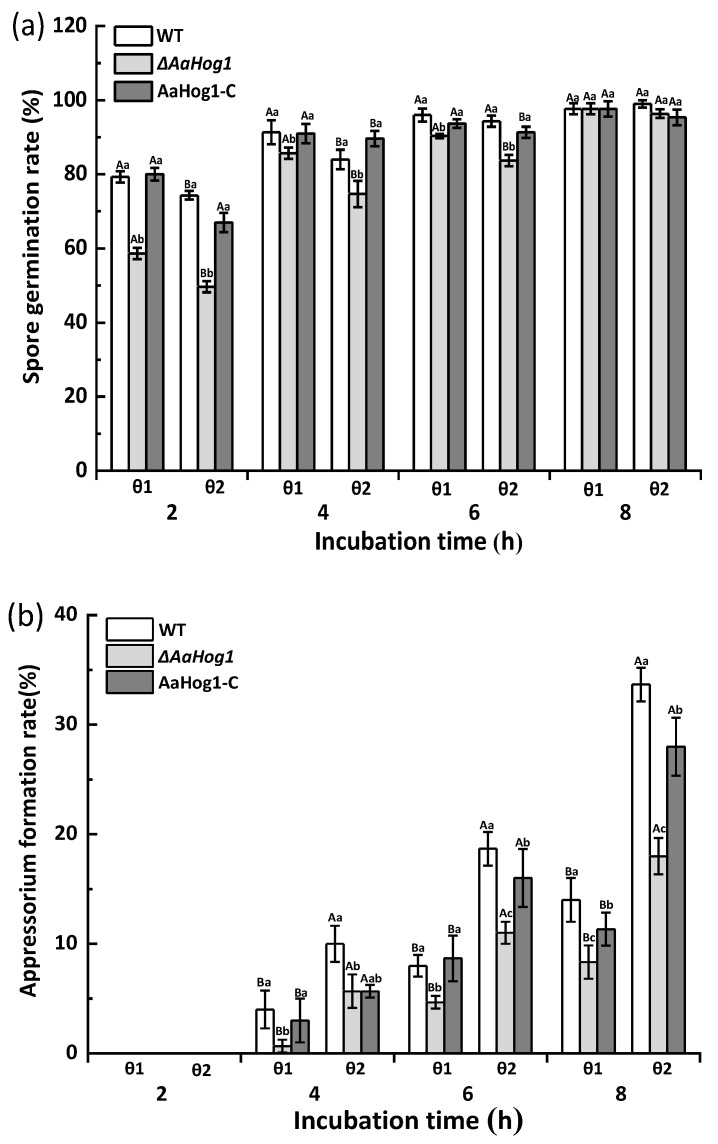
The effect of *AaHog1* on the spore germination (**a**,**c**) and appressorium formation (**b**,**d**) rates of *A. alternata* induced by different hydrophilic/hydrophobic (θ1/θ2), fruit wax (F), paraffin (P), and beeswax (B) surfaces. The effect of *AaHog1* on the appressorium and infection hyphae (**e**,**f**) formation rates of *A. alternata* on the surface of intact (θ3), dewaxed (θ4), fruit-wax-extract-coated (θ5), and fruit-wax-extract-coated dewaxed (θ6) onion epidermises. The vertical line indicates the standard error (±SE). According to Duncan’s multiple range test, uppercase letters indicate the significant differences between groups; lowercase letters indicate the significant differences between WT, Δ*AaHog1* mutant, and AaHog1-c strains (*p* < 0.05).

**Figure 3 jof-08-00266-f003:**
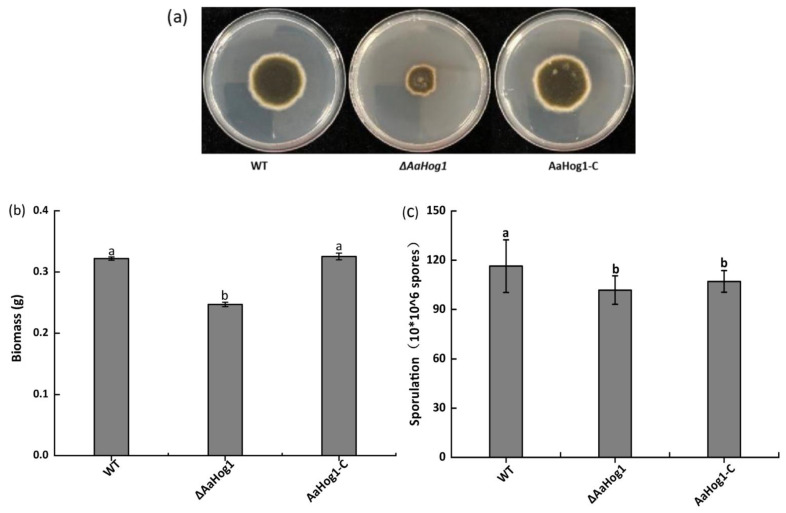
The effect of AaHog1 on hyphal growth (**a**), biomass (**b**), and sporulation (**c**) of *A. alternata*. According to Duncan’s multiple range test, treatments followed by different letters are significantly different (*p* < 0.05).

**Figure 4 jof-08-00266-f004:**
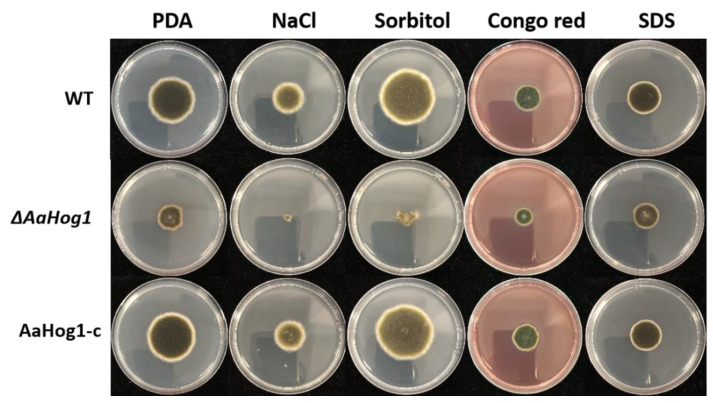
Stress tolerance assays of wild-type (WT), Δ*AaHog1* mutant, and AaHog1 genetic complementation (AaHog1-c) strains. WT, Δ*AaHog1*, and AaHog1-c were treated with different chemical components in PDA. Pictures were taken after six days of incubation on PDA containing 1 mol L^−1^ NaCl, 1 mol^−1^ sorbitol, and 10 μmol^−1^ Congo red or 1 mol^−1^ sodium dodecyl sulfate (SDS).

**Figure 5 jof-08-00266-f005:**
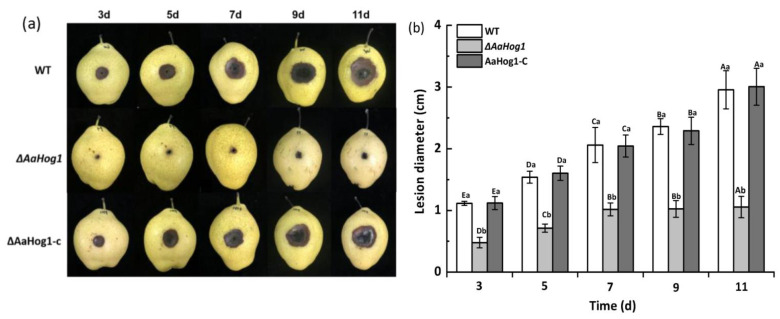
Virulence assays of the WT, Δ*AaHog1* mutant, and *AaHog1* genetic complementation strains (AaHog1-c). Each wound site (2 mm deep, 5 mm wide) of ‘Zaosu’ pear fruits was inoculated with 20 μL conidial suspensions (1 × 10^6^ spores mL^−1^) of WT, Δ*AaHog1* mutant, or AaHog1-c strains. Nine pear fruits were inoculated using each strain. The inoculated fruits were packed in plastic boxes, stored at 23 °C under 55% humidity, and observed for disease symptoms (**a**). Lesion diameters of WT, Δ*AaHog1* mutant, or AaHog1-c strains (**b**). The bars indicate standard errors (±SE). According to Duncan’s multiple range test, uppercase letters indicate significant differences between different times; lowercase letters indicate the significant differences between WT, Δ*AaHog1* mutant, and AaHog1-c strains (*p* < 0.05).

**Figure 6 jof-08-00266-f006:**
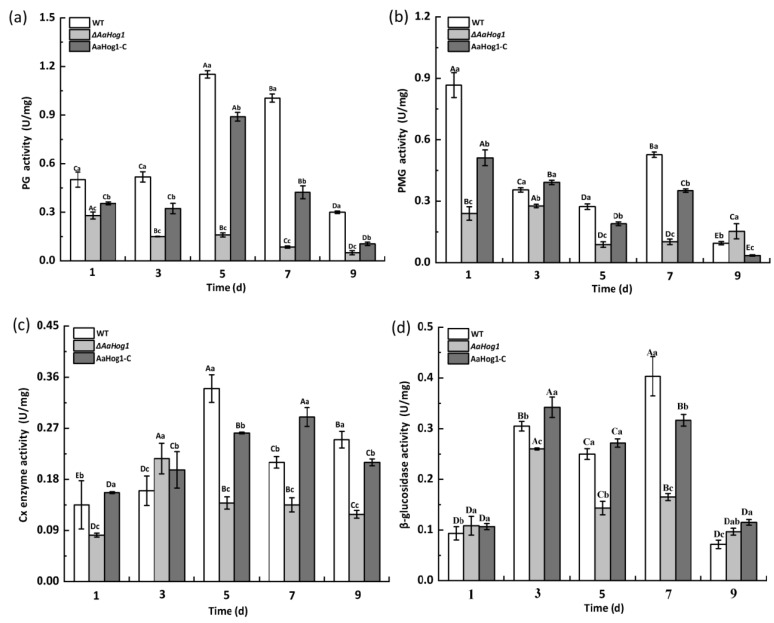
The effect of *AaHog1* on polygalacturonase (PG) (**a**), pectin methylesterase (PMG) (**b**), cellulase (Cx) (**c**), and β-glucosidase (**d**) activities. The bars indicate standard errors (±SE). According to Duncan’s multiple range test, uppercase letters indicate significant differences between different times; lowercase letters indicate the significant differences between WT, Δ*AaHog1* mutant, and AaHog1-c strains (*p* < 0.05).

**Table 1 jof-08-00266-t001:** Percentage of Δ*AaHog1* and AaHog1-c mycelial inhibition under exogenous stress. Uppercase letters indicate sig-nificant differences between groups (*p* < 0.05).

Treatment	Percentage of Inhibition (%)		
	WT	Δ*AaHog1*	AaHog1-c
PDA	-	40.31 ± 0.02 ^A^	0.20 ± 0.05 ^B^
1 mol L^−1^ NaCl	41.35 ± 0.1 ^B^	87.25 ± 0.02 ^A^	44.21 ± 0.03 ^B^
1 mol^−1^ Sorbitol	−32.08 ± 0.06 ^C^	66.76 ± 0.01 ^A^	−28.42 ± 0.06 ^B^
10 μmol^−1^ Congo red	49.32 ± 0.08 ^B^	64.75 ± 0.03 ^A^	51.47 ± 0.07 ^B^
1 mol^−1^ SDS	36.33 ± 0.06 ^B^	51.55 ± 0.03 ^A^	38.19 ± 0.06 ^B^

## Data Availability

The data presented in this study are available upon request from the corresponding authors.

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
