# Peer review of "AaHog1 Regulates Infective Structural Differentiation Mediated by Physicochemical Signals from Pear Fruit Cuticular Wax, Stress Response, and Alternaria alternata Pathogenicity"

_jof, 2022, doi:10.3390/jof8030266_

Round 1

Reviewer 1 Report

In this research, the function of AaHog1 during the appressorium and infection hyphae formation of A. alternata induced by physicochemical signals of pear fruit cuticular wax were assessed. Moreover, the role of of the AaHog1 on stress response, pathogenicity and cell wall degrading enzyme activity of A. alternata was investigated.

The manuscript shows a complete work, well-contextualized and well-discussed, being the results of interest for scientific community. Only a minor comment:

  • Line 75-80. The paragraph “In general, our results showed that the…………….. on pathogenic of A. alternata.” should be removed.

Author Response

Thank you very much for giving us such good suggestions about revising this paper. According to your suggestions and comments, we have earnestly revised this paper. The details of the changes made during revision are as follow:

Point 1: Line 75-80. The paragraph “In general, our results showed that the…………….. on pathogenic of A. alternata.”should be removed. 

Response 1: As the reviewer’ s suggestion, the paragraph “In general, our results showed that the…………….. on pathogenic of A. alternata." has been removed. 

Reviewer 2 Report

In this manuscript, Zhang and colleagues set out to characterize the phenotypes of a ∆Hog1 in the fungal pathogen Alternaria alternata. The authors also produced a complementation strain ("AaHog1-C") to verify the revision of the phenotypes of the ∆Hog1 strain. The rationale is very straightforward (that is, the whole report is a characterization of a deletion strain) and could be of interest for an audience interested in plant pathology. Nevertheless, I found that the text is not supported by the data in many instances. Also, the way the data is reported is not always explicit and difficult to match with the graphs. It could be that it is my fault for not interpreting correctly the manuscript, so I would suggest a "major revision" rather than "rejection" of this manuscript. My concerns and comments are detailed below:

The manuscript requires an overall linguistics revision. I found several typos and grammar mistakes that I will not list here. For example, "Conge Red" is wrongly used throughout the manuscript; the correct name of this compound is Congo Red. Another example: in line 188, the verbal tense is wrong (the present is used, while past tense is used in the remaining sections of the Methods. These are just two examples. There are other language problems.

The Introduction offers an adequate review of the related literature.

Method description is insufficient. For example, the molecular cloning procedure is not explained in detail. There is no mention of GFP in the Methods, nor the regulatory sequences (like promoter and/or terminator) or restriction sites used.

Statistics are missing in Fig. 1B

In Fig. 1C, the quality (resolution) of the microscopy is very poor. The DAPI channel seems to show a total overlap with GFP. Other problems seem to be related to the saturation levels of the images. For example, at 4hrs, the whole cell on the right side is 100% DAPI positive. It is not reasonable to think that the nuclei of this cell occupy 100% of the area. There are similar problems in other panels.

line 247: it seems to me that both hydrophilic and hydrophobic surfaces stimulated appressorium formation, in contrast to what is stated in the text

In Fig.2B/8 hours, it is difficult to understand how it is possible, looking at the bars, that hydrophilic (01) and hydrophobic (02) conditions are in the same statistical group. In several cases, statistical grouping based on ANOVA seems to be incorrectly showing the same group for conditions with clearly different yy axis values.

Line 251: "11% and 94%": I don't understand how this was calculated; seems to not match the graph

Line 254: I do not see differences in spore germination in wild type Alternaria alternata grown on F vs. P or B. There is a difference in appressorium formation but not "dramatic" as stated.

Line 264: What do the authors mean by "by contrast"? By contrast to...?

Lines 285/286: Statement is not fully supported by the data. The AaHog1-C show similar defect in sporulation as the ∆Hog1 strain.

Line 290: "Growth inhibitory rate" is a misleading phrase. Why not simply use "Percentage of inhibition"?

Table 1: The meaning of "A", "B" and "C" is not clearly explained. Statistical groups?

Line 342: The expression "in most cases" is not acceptable. Differences between conditions should be explicitly reported.

Author Response

Thank you very much for giving us such good suggestions about revising this paper. According to your suggestions and comments, we have earnestly revised this paper. The details of the changes made during revision are as follow:

Point 1: The manuscript requires an overall linguistics revision. I found several typos and grammar mistakes that I will not list here. For example, "Conge Red" is wrongly used throughout the manuscript; the correct name of this compound is Congo Red. Another example: in line 188, the verbal tense is wrong (the present is used, while past tense is used in the remaining sections of the Methods. These are just two examples. There are other language problems.

Response 1: As the reviewer‘s suggestion, we totally agree and the manuscript has been revised and re-polished by a native English speakers.

Point 2: The Introduction offers an adequate review of the related literature.

Response 2: Thank you for your careful review.

Point 3: Method description is insufficient. For example, the molecular cloning procedure is not explained in detail. There is no mention of GFP in the Methods, nor the regulatory sequences (like promoter and/or terminator) or restriction sites used.

Response 3: As the reviewer‘s suggestion, we have checked and provided detailed procedure in M&M section..

Point 4: Statistics are missing in Fig. 1B

Response 4: Thank you for your careful review, we have checked and supplemented

Point 5: In Fig. 1C, the quality (resolution) of the microscopy is very poor. The DAPI channel seems to show a total overlap with GFP. Other problems seem to be related to the saturation levels of the images. For example, at 4hrs, the whole cell on the right side is 100% DAPI positive. It is not reasonable to think that the nuclei of this cell occupy 100% of the area. There are similar problems in other panels.

Response 5: As the reviewer’s comment, the quality (resolution) of the picture has been improved as possible. Higher level DAPI positive is partially is due to multinuclear of A. alternata.  

Point 6: line 247: it seems to me that both hydrophilic and hydrophobic surfaces stimulated appressorium formation, in contrast to what is stated in the text

Response 6: As the reviewer’s comment, our explanation is that only a small amount of appressorium was formed on the hydrophilic surface (θ1). In comparison with hydrophilic surface, the appressorium formation rate of A. alternata was dramatically induced by hydrophobic film (θ2) surface (Fig.2a/b).

Point 7: In Fig.2B/8 hours, it is difficult to understand how it is possible, looking at the bars, that hydrophilic (θ1) and hydrophobic (θ2) conditions are in the same statistical group. In several cases, statistical grouping based on ANOVA seems to be incorrectly showing the same group for conditions with clearly different yy axis values.

Response 7: As the reviewer’s comment, we totally agree and re-added statistics. Our explanation is that Fig.2B is appressorium formation rates of WT, ∆AaHog1 mutant and AaHog1-c strains in hydrophilic surface (θ1) and hydrophobic (θ2) conditions.

Point 8: "11% and 94%": I don't understand how this was calculated; seems to not match the graph

Response 8: Thank you for your careful review. In comparison with the WT strains, the spore germination and appressorium formation rate of the ∆AaHog1 mutant strain on hydrophobic film surfaces were reduced by 11% and 94% after 4 h of incubation, respectively. Reduce rates=((A-B)/A)*100%, A represent the spore germination and appressorium formation rate of WT strain, B represent the spore germination and appressorium formation rate of ∆AaHog1 mutant strain.

Point 9: I do not see differences in spore germination in wild type Alternaria alternata grown on F vs. P or B. There is a difference in appressorium formation but not "dramatic" as stated.

Response 9: Thank you for your careful review. As shown in Figure 2c/d, In comparison with the paraffin (P) and beeswax (B), fruit wax (F) significantly induced the spore germination (2 h) and appressorium formation (4h-) rates of A. alternata.

Point 10: Line 264: What do the authors mean by "by contrast"? By contrast to...?

Response 10: As the reviewer’s comment, "By contrast" is not appropriate here, we totally agree and have deleted it.

Point 11: Lines 285/286: Statement is not fully supported by the data. The AaHog1-C show similar defect in sporulation as the ∆Hog1 strain.

Response 11: As the reviewer’s comment, the AaHog1-c strain only partially recovered the defect in sporulation of the ∆AaHog1 mutant strains.

Point 12: Line 290: "Growth inhibitory rate" is a misleading phrase. Why not simply use "Percentage of inhibition"?

Response 12: As the reviewer’s comment, we totally agree and revised it.

Point 13: Table 1: The meaning of "A", "B" and "C" is not clearly explained. Statistical groups?

Response 13: As the reviewer’s comment, we have added corresponding table notes.

Point 14: Line 342: The expression "in most cases" is not acceptable. Differences between conditions should be explicitly reported.

Response 14: As the reviewer’s comment, we totally agree and revised it.

Reviewer 3 Report

General comments

Change article title to

The role of AaHog1 in the regulation of infective structure differentiation mediated by physicochemical signals from pear fruit cuticular wax, stress response, and pathogenicity of Alternaria alternata

Fig. 1.A. requires a resolution improvement.

More Updated review literature is needed in introduction and discussion

Make as many typographical corrections as possible.

Many sentences do not have the correct punctuation, and it is difficult to read the text.

English should be improved; grammar needs enhancement in many sentences and paragraphs.

The conclusion you have provided is quite brief and provides sufficient feedback on the main objectives of your study.

Author Response

Thank you very much for giving us such good suggestions about revising this paper. According to your suggestions and comments, we have earnestly revised this paper. The details of the changes made during revision are as follow:

Point 1: Change article title to:The role of AaHog1 in the regulation of infective structure differentiation mediated by physicochemical signals from pear fruit cuticular wax, stress response, and pathogenicity of Alternaria alternata

Response 1: As the reviewer’s suggestion, we totally agree and have changed as: The role of AaHog1 in the regulation of infective structure differentiation mediated by physicochemical signals from pear fruit cuticular wax, stress response, and pathogenicity of Alternaria alternata.

Point 2: Fig. 1.A. requires a resolution improvement.

Response 2: As the reviewer’s suggestion, we have provided the higher resolution Figure 1A.

Point3: More Updated review literature is needed in introduction and discussion

Response 3: As the reviewer’s suggestion, some updated review literature have been added in Introduction and Discussion section.

Point4: Make as many typographical corrections as possible.

Response 4: As the reviewer’s suggestion, we have carefully checked and revised.

Point5: Many sentences do not have the correct punctuation, and it is difficult to read the text.

Response 5: Thank you for your careful review, we have carefully checked and revised.

Point6: English should be improved; grammar needs enhancement in many sentences and paragraphs.

Response 6: As the reviewer’s comment, the manuscript has been revised and re-polished by a native English speaker.

Point7: The conclusion you have provided is quite brief and provides sufficient feedback on the main objectives of your study.

Response 7: Thank you for your careful review. The conclusion of manuscript has been further revised.

Reviewer 4 Report

In my opinion, the results presented in this manuscript are very interesting. Overall, the experimental design and data analysis are appropriate, and the introduction and discussion are consistent enough. However, I have minor comments (mainly technical suggestions) on the manuscript prior to its official approval for printing.

My comments
1) Fig 1. “...Scale bar=20 micron....” ?

Please add scales to each photo and describe it correctly.

2) Tables 1 – Please describe what the individual data in the table mean, such as SD, statistics, etc. Please do it in the same way as, for example, in the case of a Fig. 2.

3) Discussion – why are the literature references blue? In my opinion, this section should be described in more detail.

Author Response

Thank you very much for giving us such good suggestions about revising this paper. According to your suggestions and comments, we have earnestly revised this paper. The details of the changes made during revision are as follow:

Point 1: Fig 1. “...Scale bar=20 micron....” ?

Please add scales to each photo and describe it correctly.

Response 1: Thank you for your careful review, we have added scales to each photo.

Point 2: Tables 1-Please describe what the individual data in the table mean, such as SD, statistics, etc. Please do it in the same way as, for example, in the case of a Fig. 2.

Response 2: As the reviewer’ s comment, we have added corresponding table notes.

Point 3: Discussion- why are the literature references blue? In my opinion, this section should be described in more detail.

Response 3: Thanks for the reviewer’ s comment, according to the requirements of the journal, the literature references in manuscript is blue; in addition the Discussion section have been further revised.

Round 2

Reviewer 2 Report

I appreciate and recognize the efforts undertook by Zhang and colleagues to improve the quality of the manuscript. Nevertheless, I still think that the conclusions are not supported by the data and my comments have not been properly addressed.

While a grammar revision was performed, the manuscript still requires several language-related edits. Examples (there are many more): in line 186, "was added test tubes"; in line 205, should be "contains".

The Methods are improved but still insufficient. Vector construction remains incomplete. The plasmid backbone is now indicated but no references or indication of what is the promoter driving the expression of the target gene nor if there were other modifiers such as a terminator.

I think statistics are not well indicated in the figures. The letters should indicate "statistical groups", that is, all samples with the same letter should not show statistical differences amongst them. In Fig. 1B, how can the bars for 2hrs and the bar for 4 hrs (+ wax) both have "a"? They are clearly different. This is just an example.

Multinucleate cells do not justify the signal saturation in Fig. 1C. The DAPI fluorescence occupies 100% of the cell in some instances of the figure. It does not seem biologically plausible that the nuclei are occupying 100% of the cell's area; instead, I believe this is a technical artifact.

In some other comments made by the authors, it appears like they acknowledge the problem indicated in my criticisms but did not substantially improve the text.

In summary, I think that despite the laudable efforts put in by the authors, the text is not fully supported by the data so I am unable to recommend this manuscript for publication.

Author Response

Point 1: While a grammar revision was performed, the manuscript still requires several language-related edits. Examples (there are many more): in line 186, "was added test tubes"; in line 205, should be "contains".

Response 1: As the reviewer‘s suggestion, we totally agree and the manuscript has been edited by a native English-speaking editor of MogoEdit.

Point 2: The Methods are improved but still insufficient. Vector construction remains incomplete. The plasmid backbone is now indicated but no references or indication of what is the promoter driving the expression of the target gene nor if there were other modifiers such as a terminator.

Response 2: Thank you for your careful review, we apologize for not describing the method clearer. The AaHog1 complementation strain (AaHog1-c) were constructed in reference to the method described previously Chen et al., 2018; Li et al., 2021. A 1.3 Kb DNA fragment containing the entire AaHog1 gene as well as the promoter region Olic and terminator region Trpc were cloned into pC-NEO-CGFP…, We have now supplemented this in the Methods.

Point 3: I think statistics are not well indicated in the figures. The letters should indicate "statistical groups", that is, all samples with the same letter should not show statistical differences amongst them. In Fig. 1B, how can the bars for 2hrs and the bar for 4 hrs (+ wax) both have "a"? They are clearly different. This is just an example.

Response 3: Thank you for your careful review. Significant differences are relabeled in Figure 1B and Figure 5B. In Fig2/5/6, uppercase letters indicate differences between groups; lowercase letters indicate the differences within groups (different strains) in manuscript (P < 0.05).

Point 4: Multinucleate cells do not justify the signal saturation in Fig. 1C. The DAPI fluorescence occupies 100% of the cell in some instances of the figure. It does not seem biologically plausible that the nuclei are occupying 100% of the cell's area; instead, I believe this is a technical artifact.

Response 4: Thank you very much for giving us such good suggestions, we have repeated the subcellular localization experiments. 100% DAPI positive at 4hrs in previous figures is mainly due to a technical issue. We have supplemented the new results.

Round 3

Reviewer 2 Report

Thank you for the effort in submitting a revised version.

Still some linguistic mistakes but I guess these could be revised by the journal’s office editors. Some examples:

Line 22: “pathogenic of A. alternata” should be “pathogenesis of A. alternata”

Line 46-48 requires revision, words missing?

Line 93: “was” instead of “were”

Line 136, line 148: “Essentially”??? Methods should be described in full, not only the “essential” parts

It could be that I may be lacking a correct understanding, but the indication of statistical groups in bar charts seems to remain wrong (Fig. 1B, 2, 5B and 6). I suggest the authors to obtain detailed information on how to present statistical analyses of ANOVA followed by a post-hoc test such as Duncan’s.

I appreciate the effort made by the authors to update the fluorescence microscopy shown on Fig. 1C. However, I still think that signal saturation is causing the authors to make potentially wrong conclusions. The authors should try and make an effort to present more convincing data. For example, please show technical controls and unsaturated images. Actually, a search on the localization of Hog1 in fungal cells revealed that the localization of AaHog1 had been previously reported (see Lin and Chung, 2010, Fungal Genetics and Biology).

The reference (Lin and Chung, 2010, Fungal Genetics and Biology), whose title is “Specialized and shared functions of the histidine kinase- and HOG1 MAP kinase-mediated signaling pathways in Alternaria alternata, a filamentous fungal pathogen of citrus” reports the localization of Hog1 and some phenotypes of the deletion of AaHog1 of Alternaria alternata, including sensitivity of the deletion strain to exogenous stresses. This article is only briefly mentioned in the manuscript currently under review. The data present in the published report and the current manuscript should be at least compared. Given the relevance of the previously research for the very specific topic under investigation in the manuscript submitted, it seems an important aspect missing in the manuscript. It also removes some of the novelty of the present article. I was not aware of Lin and Chung, 2010 when I wrote my previous two reviews of this manuscript.

Author Response

Thank you very much for giving us such good suggestions about revising this paper. According to your suggestions and comments, we have earnestly revised this paper. The details of the changes made during revision are as follow:

Point1: Line 22: “pathogenic of A. alternata” should be “pathogenesis of A. alternata

As the reviewer’s suggestion, we have carefully checked and revised.

Point2: Line 46-48 requires revision, words missing?

As the reviewer’s suggestion, we have carefully checked and revised.

Point3: Line 93: “was” instead of “were”

As the reviewer’s suggestion, we have carefully checked and revised.

Point4: Line 136, line 148: “Essentially”??? Methods should be described in full, not only the “essential” parts

As the reviewer’s suggestion, we have carefully checked and revised.

Point5: It could be that I may be lacking a correct understanding, but the indication of statistical groups in bar charts seems to remain wrong (Fig. 1B, 2, 5B and 6). I suggest the authors to obtain detailed information on how to present statistical analyses of ANOVA followed by a post-hoc test such as Duncan’s.

Thank you for your careful review. For statistical analysis, Duncan's test was carried out to determine the significance of difference. For example Fig. 2A/B: uppercase letters indicate the significant differences between hydrophilic and hydrophobic films surface at the same incubation time. lowercase letters indicate the significant differences between WT, ΔAaHog1 mutant and AaHog1-c strains at the same incubation time.

Point6: I appreciate the effort made by the authors to update the fluorescence microscopy shown on Fig. 1C. However, I still think that signal saturation is causing the authors to make potentially wrong conclusions. The authors should try and make an effort to present more convincing data. For example, please show technical controls and unsaturated images. Actually, a search on the localization of Hog1 in fungal cells revealed that the localization of AaHog1 had been previously reported (see Lin and Chung, 2010, Fungal Genetics and Biology).

After our considerable discussion, we accept the reviewer's doubt. We decided to delete the result of subcellular localization, which does not affect the functional validation of Hog1. We will try to improve the technology of subcellular localization in the future.

Point7: The reference (Lin and Chung, 2010, Fungal Genetics and Biology), whose title is “Specialized and shared functions of the histidine kinase- and HOG1 MAP kinase-mediated signaling pathways in Alternaria alternata, a filamentous fungal pathogen of citrus”reports the localization of Hog1 and some phenotypes of the deletion of AaHog1 of Alternaria alternata, including sensitivity of the deletion strain to exogenous stresses. This article is only briefly mentioned in the manuscript currently under review. The data present in the published report and the current manuscript should be at least compared. Given the relevance of the previously research for the very specific topic under investigation in the manuscript submitted, it seems an important aspect missing in the manuscript. It also removes some of the novelty of the present article. I was not aware of Lin and Chung, 2010 when I wrote my previous two reviews of this manuscript.

Thank you for your careful review. Lin and Chung, 2010 showed very good results in the partial functional verification of the Hog1 gene, especially the subcellular localization technology and image presentation are worth learning. However, our study focused on the role of AaHog1 in the regulation of infective structure differentiation mediated by physicochemical signals from pear fruit cuticular wax. While Lin reported that Hog1 genes in cellular responses to osmotic and oxidative stresses, sensitivity to fungicides, and resistance to multidrugs in this major citrus pathogen. Previously, we have found that the physicochemical signals from pear fruit cuticular wax can induce the infective structures formation of A. alternata (Li, et al., 2008; Tang, et al., 2017;) After our research over the years, we found that the Ga2+ signal and cAMP-PKA signaling pathway are involved in the  infective structures formation of A. alternata induced by physicochemical signals from pear fruit cuticular wax. We will further study the cross-talk between these signaling pathways . In addition, our results further explored the pathogenic mechanism of Hog1 affecting A. alternata. I believe our results may be useful for dissecting the regulatory mechanism of AaHog1 MAP kinase on pathogenic of A. alternata, the causal agent of black spot in pear fruit.
